# The Cardiometabolic Impact of Rebaudioside A Exposure during the Reproductive Stage

**DOI:** 10.3390/biology13030163

**Published:** 2024-03-02

**Authors:** Isabella Bracchi, Juliana Morais, João Almeida Coelho, Ana Filipa Ferreira, Inês Alves, Cláudia Mendes, Beatriz Correia, Alexandre Gonçalves, João Tiago Guimarães, Iněs Falcão-Pires, Elisa Keating, Rita Negrão

**Affiliations:** 1Unit of Biochemistry, Department Biomedicine, Faculty of Medicine, University of Porto, 4200-319 Porto, Portugal; isabellabracchi@gmail.com (I.B.); jtguimar@med.up.pt (J.T.G.); keating@med.up.pt (E.K.); 2CINTESIS, Center for Health Technology and Services Research, 4200-319 Porto, Portugal; ju_morais17@hotmail.com; 3Department of Functional Sciences, School of Health, Polytechnic Institute of Porto, 4200-072 Porto, Portugal; 4Department of Surgery and Physiology, Faculdade de Medicina, Universidade do Porto, 4200-319 Porto, Portugal; almeidacoelho.joao@gmail.com (J.A.C.); anaferreira.ferreira4@gmail.com (A.F.F.); minalves09@gmail.com (I.A.); csmendes@med.up.pt (C.M.); beatrizmsc000@gmail.com (B.C.); alexandregoncalvs@gmail.com (A.G.); 5UniC@RISE, Unidade de Investigação e Desenvolvimento Cardiovascular, Faculdade de Medicina, Universidade do Porto, 4200-319 Porto, Portugal; 6Nutrition & Metabolism, NOVA Medical School|FCM, NOVA University Lisbon, 1169-056 Lisbon, Portugal; 7Clinical Pathology, São João University Hospital Center, 4200-319 Porto, Portugal; 8EPIUnit, Institute of Public Health, University of Porto, 4200-319 Porto, Portugal; 9CINTESIS@RISE, Faculty of Medicine, University of Porto, 4200-319 Porto, Portugal

**Keywords:** rebaudioside A, pregnancy, lactation, cardiometabolism

## Abstract

**Simple Summary:**

Since foods rich in refined sugars promote obesity, the use of non-caloric sweeteners has gained popularity, and their consumption by pregnant women has increased. Stevia (a non-caloric sweetener) consumption was considered safe for humans by the European Food Safety Authority in a dose of up to 4 mg/kg body weight/day. However, the World Health Organization recommended in 2023 the restraint of these sweeteners at any life stage, highlighting the need for research on pregnant women and early stages of development. So, we aimed to study the effects of chronic consumption of the main sweetener compound of stevia (Rebaudioside A) during the reproductive stage. Female rats were treated with Rebaudioside A (4 mg steviol equivalents/kg body weight/day) in the drinking water from 4 weeks before mating until weaning. Food and water consumption, blood glucose and lipids, as well as heart structure, function and mitochondrial function, were assessed. Rebaudioside A decreased heart size, cardiomyocyte area and fibrosis without repercussions on cardiac or mitochondrial function. Both fasting blood glucose and cholesterol decreased. This work suggests that stevia consumption at this dose may be safe for females during the reproductive stage. However, more studies are mandatory to explore the effects of stevia consumption on offspring’s health.

**Abstract:**

The consumption of non-sugar sweeteners (NSS) has increased during pregnancy. The European Food Safety Agency suggested that steviol glycosides, such as Rebaudioside A (RebA), the major sweetener component of stevia, are safe for humans up to a dose of 4 mg/kg body weight/day. However, the World Health Organization recommended in 2023 the restraint of using NSS, including stevia, at any life stage, highlighting the need to study NSS safety in early periods of development. We aimed to study the mitochondrial and cardiometabolic effects of long-term RebA consumption during the reproductive stage of the life cycle. Female rats were exposed to RebA (4 mg steviol equivalents/kg body weight/day) in the drinking water from 4 weeks before mating until weaning. Morphometry, food and water consumption, glucose and lipid homeostasis, heart structure, function, and mitochondrial function were assessed. RebA showed an atrophic effect in the heart, decreasing cardiomyocyte cross-sectional area and myocardial fibrosis without repercussions on cardiac function. Mitochondrial and myofilamentary functions were not altered. Glucose tolerance and insulin sensitivity were not affected, but fasting glycemia and total plasma cholesterol decreased. This work suggests that this RebA dose is safe for female consumption during the reproductive stage, from a cardiometabolic perspective. However, studies on the effects of RebA exposure on the offspring are mandatory.

## 1. Introduction

Obesity is a multifactorial and complex disease and one of the most common risk factors for cardiovascular disease. Obesity prevalence is alarmingly high worldwide. The World Health Organization (WHO) European Regional Obesity Report states that obesity affected 23.3% of the adult population in the WHO European region in 2022 [1]. The report also estimates that more than 20% of European women have obesity before pregnancy [1].

Since excessive energy intake from foods rich in free sugars (such as sucrose, glucose, or fructose) promotes overweight and obesity [2], the use of non-sugar sweeteners (NSS) as low-calorie alternatives has gained popularity in recent decades as a means of preventing obesity. In fact, the USA authorities reported a 50% increase in the prevalence of low-calorie sweeteners consumption by pregnant women from 1999 to 2014 [3].

Stevia is a generic name attributed to the plant, leaves, and sweetener compounds of *Stevia rebaudiana* Bertoni [4]. It is a natural NSS, introduced in the market as a sweetener in the 1970s [4]. The sweet-tasting properties of stevia are due to steviol glycosides, of which rebaudioside A (RebA) is the major component with 200 to 400 times sweetening capacity compared with sucrose [4].

Paradoxically, the consumption of NSS has been associated with increased food intake, weight gain, and adiposity in animal and human studies, suggestive of increased cardiometabolic risk and altered gut microbiota [5,6].

Not surprisingly, this topic has raised controversy. The European Food Safety Authority (EFSA) Panel on Food Additive and Flavourings (FAF) concluded in 2021 that using steviol glycosides up to 4 mg steviol equivalents/kg body weight/day presented no safety concerns for humans [7]. On the other hand, the WHO released in May 2023 a guideline recommending the restraint of using NSS, including stevia, for weight gain control at any life stage [8]. Importantly, this guideline highlights the need for future research addressing the potential long-term effects of NSS use in children and in pregnant and lactating women.

Considering the above, this work aimed to study the cardiometabolic effects of a natural NSS long-term consumption during the reproductive stage of a rodent’s life cycle, while searching for putative mitochondrial involvement. To address this objective, 8-week-old Sprague Dawley females were exposed to 4 mg steviol equivalents/kg body weight/day /kg body weight/day RebA, the human dose corresponding to the EFSA’s acceptable daily intake (ADI) [9], in drinking water, for 13 weeks (from 4 weeks before mating, throughout pregnancy and lactation), until sacrifice at 21 weeks of age. Morphometry, food and water consumption, glucose and lipid homeostasis, and heart structure, function, and mitochondrial function were assessed.

## 2. Materials and Methods

### 2.1. Animals and Treatments

This project was approved by the Animal Welfare and Ethics Body (ORBEA) from the Faculty of Medicine of the University of Porto, Portugal, and by the Directorate General of Food and Veterinary of the Portuguese Government (0421/000/000/2022).

As depicted in Figure 1, 8-week-old female Sprague Dawley rats (Charles River Laboratories, Barcelona, Spain) were kept under controlled environmental conditions (22–24 °C and 12 h light/dark cycles) for 1 week before the beginning of the experimental treatment and thereafter. Beverage and food (were administered ad libitum during the entire experiment. Body weight, beverage, and food consumption were evaluated daily during this acclimation week and then weekly during the entire experiment. The naso-anal length was measured weekly. At 8 weeks, females were randomly distributed into two groups: (i) the RebA group (*n* = 8), which was consumed 4.71–5.61 mg steviol equivalents/kg body weight/day (according to the ADI set by the Scientific Committee on Food (SCF) and EFSA [9]) (Rebaudioside A 96% HPLC, 01432-0010, Sigma-Aldrich^®^, Lisbon, Portugal) in the drinking water starting 4 weeks before mating, throughout pregnancy, and until the end of lactation, for a total duration of 13 weeks of treatment; and (ii) the control group (Control, *n* = 8) receiving drinking water. Both groups received the same standard diet (Mmucedola_s.r.l._, 4RF21, Settimo Milanese, Italy). Animals’ weight was measured weekly, and RebA concentration was adjusted twice a week according to beverage consumption. Upon mating, elicited according to estrous cycle staging, pregnancy was confirmed by vaginal plug detection and vaginal cytology, allowing for the estimation of the delivery date. Mating efficiency (the number of encounters with males until conception) was also evaluated, as well as the gestational age (in days) at birth, litter size (*n* pups), and female-to-male ratio of each litter.

At 21 weeks of age, after weaning, female rats were sacrificed by exsanguination under anesthesia by intraperitoneal administration of 400 mg/mL sodium pentobarbital (Euthanimal, Nephar). Blood was obtained via cardiac puncture, and plasma and serum were separated and preserved at −80 °C for future analysis. Organs and tissues were dissected, weighed, and immediately frozen in liquid nitrogen and preserved at −80 °C or fixed in 10% formaldehyde (PanReac AppliChem, Castellar del Vallès, Spain) and processed until its inclusion in paraffin. 

As considered equivalent to the body mass index in humans, the Lee index was determined using the following equation: Lee index = body weight^1/3^ (g)/nasal–anal length (cm) × 1000.

Body surface area (BSA = 9.1 × body weight^.667^(g)) was calculated and used to normalize echocardiographic dimensions, mass and organ weights.

### 2.2. Oral Glucose Tolerance Assessment and Insulin Sensitivity Assessment

Oral glucose tolerance test (OGTT) was performed in 7 h fasting animals after administering glucose (2 g/kg body weight, Fisher Scientific, London, UK) by gavage. Blood glucose concentration was measured with a FreeStyle Precision Neo system (Abbott, Amadora, Portugal) at 0, 15, 30, 60, 90, and 120 min of glucose oral gavage.

Insulin sensitivity test was performed in 7 h fasting animals after an intraperitoneal injection of insulin (0.75 U/Kg body weight, Actrapid^®^, Novo Nordisk^®^, Paço de Arcos, Portugal), and blood glucose concentration was measured as mentioned above.

### 2.3. Plasma Biochemical Markers Determination

Plasma biochemical markers were measured in the Central Laboratory, Department of Clinical Pathology, Centro Hospitalar Universitário São João, using conventional methods with an AU5400 automated clinical chemistry analyzer (Beckman-Coutler, Paço de Arcos, Portugal). 

Hepatic function was evaluated by the determination of aspartate aminotransferase (AST), alanine aminotransferase (ALT), and alkaline phosphatase (ALP). Blood lipid parameters such as triglycerides, total cholesterol, and high-density lipoprotein cholesterol (HDL-c) were also measured, and low-density lipoprotein cholesterol (LDL-c) was calculated using the Friedwald equation.

### 2.4. Echocardiographic Evaluation

One day before euthanasia, each animal was anesthetized in a ventilated container (sevoflurane 5% for induction and 2.5–3% for maintaining anesthesia, Abbvie, Amadora, Portugal) to assess cardiac function. A linear 15 MHz probe (Sequoia 15L8W, UMI, Bellflower, CA, USA) and an echocardiograph Acuson Sequoia C512 (Siemens, Erlangen, Germany) were used to perform the transthoracic echocardiography. After 3 consecutive heartbeats, the recordings were averaged. M-mode was used to determine systolic and diastolic wall thickness, cavity dimensions, and transverse aortic root diameter via the parasternal short-axis view. The left ventricle (LV) mass, the ejection fraction, and the fractional shortening were calculated as previously described [10]. Mitral flow velocity tracings were obtained via pulsed-wave Doppler just above the mitral leaflets [10].

### 2.5. Mitochondrial Respiration Evaluation Using Permeabilized Cardiac Fibers

A portion of freshly isolated myocardium was immediately immersed in ice-cold BIOPS, a preservation solution composed of 10 mM Ca-EGTA buffer, 0.1 µM free calcium, 20 mM imidazole, 20 mM taurine, 50 mM K-MES, 0.5 mM DTT, 6.56 mM MgCl_2_, 5.77 mM ATP, and 15 mM phosphocreatine with a pH of 7.1. Using two pairs of sharp forceps, the fiber bundles were mechanically separated in a small, ice-cold Petri dish and then placed into individual wells of a 12-well tissue culture plate with ice-cold BIOPS until all fibers were prepared. Subsequently, fiber bundles were submersed into a well with freshly prepared saponin solution (50 µg/mL of BIOPS) and incubated for 30 min with gentle agitation. Finally, the fibers were placed into 2 mL of ice-cold MiR06 [a mitochondrial respiration medium composed of MiR05 (0.5 mM EGTA, 3 mM MgCl_2_, 60 mM lactobionic acid, 10 mM KH_2_PO_4_, 20 mM taurine, 20 mM HEPES, 110 mM D-sucrose, and 1 g/L essentially fatty acid-free bovine serum albumin (BSA) with a pH of 7.1) supplemented with 280 U/mL catalase] during 10 min on ice. Permeabilized fiber bundles were then carefully blotted on filter paper, weighed, and 1–2 mg wet weight was transferred into each chamber of the high-resolution respirometer (Oroboros Instruments, Innsbruck, Austria), containing 2 mL of MiR06 [11]. Using the high-resolution respirometer Oroboros Oxygraph-2k (Oroboros Instruments, Innsbruck, Austria), mitochondrial respiratory function was assessed. The Substrate-uncoupler-inhibitor titration (SUIT)-001 O2 pfi D002 protocol for permeabilized muscle fibers was followed at 37 °C (sequential addition of 5 mM pyruvate, 2 mM malate, 7.5 mM ADP, 10 µM cytochrome C, 0.5 µM CCCP, 10 mM glutamate, 10 mM succinate, 0.5 mM octanoylcarnitine, 0.5 µM rotenone, 10 mM glycerophosphate, 2.5 µM Antimicin A, 2 mM ascorbate, 0.5 mM TMPD, and 200 mM sodium azide). Oxygen flux rate and concentration were recorded and analyzed using DatLab software (Oroboros Datlab Version 7.0, Oroboros Instruments Innsbruck, Innsbruck, Austria). Oxygen flux was expressed in pmol·s^−1^·mg^−1^ normalized for the weight of the fiber bundle (mg). All reagents referred to in this section were the ones recommended by the Oroboros Instrument, Innsbruck, Austria (https://wiki.oroboros.at/index.php/OROBOROS_INSTRUMENTS).

### 2.6. Force Measurements in Isolated Permeabilized Cardiomyocytes

The specimen preparation protocol has been previously described [12]. In brief, LV myocardial tissue from each animal was cut into small pieces, mechanically disrupted, and incubated for 5 min in extraction solution (in mM: Na_2_ATP, 5.97; MgCl, 6.28; C_3_H_6_O_2_, 40.64; BES, 100; CaEGTA, 7; and Na_2_PCr, 14.5) supplemented with 0.5% Triton X-100 (All from Merck Millipore, Burlington, MA, USA) at room temperature, to remove membrane structures. Cardiomyocytes were washed with extraction solution by consecutive centrifugations. An isolated cardiomyocyte was attached with glue between a force transducer and an electromagnetic motor length controller (Aurora Scientific Inc. (Aurora, ON, Canada) model 403A and model 315C-I, respectively). Passive tension (PT)-sarcomere length (SL) relationships ranging between 1.8 and 2.3 μm were acquired at 15 °C, with 0.1 μm step increases. Maximal activation at pCa 4.5 was used to calculate maximal calcium-activated isometric force (Total tension, Tt) and the slack test (the cell was shortened for 1 ms to 80% of its original length). A relaxing solution (in mM: Na_2_ATP, 5.89; MgCl, 6.48; C_3_H_6_O_2_, 40.76; BES, 100; CaEGTA, 6.97; and Na_2_PCr, 14.5; all reagents were from Merck Millipore, Burlington, MA, USA), pCa 9.0, was used to determine passive tension (Tp). The cardiomyocyte cross-sectional area was used to normalize the force values to the cell’s dimensions. Data acquisition was made by the ASI 600 A program with a sampling frequency of 2 KHz.

### 2.7. Histology

The heart (at the ventricular base) was sliced and fixed in 10% formaldehyde (PanReac AppliChem, Castellar del Vallès, Spain), dehydrated in ethanol (100%, 90% and 70% *v*/*v*), cleared in xylol (Fisher Scientific, London, UK), and impregnated in paraffin (“Epredia™ Paraffin Type 6”, Epredia, Breda, The Netherlands). Five-micrometer slides were dewaxed, rehydrated, stained with Hematoxylin–Eosin (HE) (Hematoxylin H and Eosin Y 1% *v*/*v* alcoholic, Biognost, Zagreb, Croatia) to assess the cardiomyocyte area, or Picrosirius Red (Direct Red 80, Sigma-Aldrich, St. Louis, MO, USA), to assess myocardial fibrosis, and finally mounted with Entellan™ (rapid mounting medium for microscopy, Merck, Darmstadt, Germany). A photographic camera (Olympus XC30, Tokyo, Japan) coupled with an optic microscope (Leitz Wetzlar—Dialux 20, Wetzlar, Germany) was used to visualize and photograph the histological preparations. The area of 60 cardiomyocytes per animal was measured using Cell^B software serial number A5607500-74EBB51C (Olympus, Tokyo, Japan). To calculate the area of fibrosis and perivascular fibrosis, eight and five fields per animal, respectively, were photographed and analyzed using Image-Pro Pus 6 software (Media Cybernetics, Rockville, MD, USA).

### 2.8. Biomarker of Myocardial Fibrotic Assessment

Transforming growth factor-beta (TGF-β1), a powerful fibrogenic cytokine, was tested by ELISA following the standard protocol provided by Elabscience Corporation (Wuhan, China).

### 2.9. Statistical Analysis

Results are expressed as mean ± SEM. Statistical analysis was performed using GraphPad Prism software (Version 10.1.2). Shapiro–Wilk was used to assess parametric distribution. Comparison between the two groups was assessed by t-test. Comparisons between more than two groups were performed by one- or two-way ANOVA, and appropriate post hoc tests were used. The probability values < 0.05 were considered significant.

## 3. Results

### 3.1. Gestational and Morphometric Data

RebA did not affect mating efficiency, gestational age at birth, litter size, or female-to-male ratio of each litter (Table 1).

Figure 2 displays weight evolution as well as the daily amount of food and water intake. RebA did not affect weight gain or the Lee index until sacrifice (Figure 2A,B). Accordingly, RebA did not significantly change water or food consumption over time (Figure 2C,D), even though the RebA group drank more water at two time points after gestation. 

Interestingly, RebA consumption reduced liver and heart weight, even after adjusting for BSA, as seen in Table 2.

### 3.2. Glycemic Control

At the end of the experiment (20 weeks of age), a significant reduction in fasting glycemia in the RebA group was observed when compared to the Control group (72.8 ± 2.18 mg/dL versus 84.8 ± 3.04 mg/dL, Figure 3A). Nevertheless, RebA chronic consumption did not affect oral glucose tolerance (Figure 3B,C) or insulin sensitivity (Figure 3D).

Two-way ANOVA indicated a significant effect of time upon all measures (*p* < 0.001) (Figure 3B,D) but revealed no statistically significant effect of RebA treatment or interaction between treatment and time for any measured parameter. 

### 3.3. Plasma Biochemical Markers 

Figure 4 shows the plasma levels of triglycerides, total cholesterol, LDL-c, and HDL-c of all animals at 21 weeks of age. RebA group presented a significantly lower plasma concentration of total cholesterol (67.75 ± 2.46 mg/dL), LDL-c (27.88 ± 1.81 mg/dL) and HDL-c (43.63 ± 1.40 mg/dL) compared to control group (total cholesterol, 84.13 ± 6.04 mg/dL; LDL-c, 35.75 ± 3.11 mg/dL and HDL-c, 53.88 ± 4.09 mg/dL). Nevertheless, the ratio HDL-c/LDL-c was not different between the two groups (1.53 ± 0.05 mg/dL for Control and 1.60 ± 0.08 mg/dL for RebA, *p* = 0.473). Regarding triglyceride levels, no differences were found between groups.

When evaluating the plasma activity of aspartate aminotransferase (AST), alanine aminotransferase (ALT), and alkaline phosphatase (ALP) at 21 weeks of age, no differences were found between groups. 

### 3.4. Left Ventricle Cardiac Structure and Functional Characterization

The echocardiographic evaluation showed no differences in cardiac function between groups. Regarding structure, posterior LV wall thickness and relative wall thickness were significantly reduced in the RebA group (25% and 12% reductions compared to control, respectively, Table 3).

Subsequently, we aimed to explore if the slight cardiac atrophy assessed by decreased heart weight/BSA (Table 2) and by echocardiography (Table 3) had any cardiac functional impact. In vivo echocardiography further confirmed that this atrophic effect had no major impact on systolic or diastolic function (Table 3). Consistently, we showed that RebA reduced cardiomyocyte cross-sectional area (Control, 358.6 ± 7.486 vs. RebA, 288.5 ± 9.381 μm^2^) (Figure 5A), cardiac interstitial fibrosis (Control, 3.99 ± 0.18 vs. RebA, 3.25 ± 0.13%) (Figure 5B), and perivascular fibrosis (Control, 18.29 ± 0.933 vs. RebA, 11.79 ± 0.583 μm^2^) (Figure 5C).

In fact, despite the differences found in the values of cardiomyocyte fibrosis and perivascular fibrosis between groups, plasma levels of TGF-β1, a generally accepted promoter of myocardial fibrosis, was similar between groups (Figure 6).

The mechanical properties of single-skinned cardiomyocytes were measured to assess function at the cardiomyocyte level without interferences from the extracellular matrix (Figure 7A). Stiffness was similar between groups, as confirmed by similar passive tension–sarcomere length relationship curves and passive tension at 2.2 μm (Figure 7B,D). Additionally, no changes were observed in the maximal developed tension or myofilament Ca^2+^ sensitivity (Figure 7C,E).

The impact of RebA administration on cardiac mitochondrial respiratory function was evaluated by high-resolution respirometry of permeabilized cardiac fibers. RebA administration did not significantly affect mitochondrial respiratory function. Namely, no differences were observed in mitochondrial oxygen fluxes using Complex I and II substrates (Figure 8). Similarly, mitochondrial outer membrane integrity and the stimulation of respiration by fatty acid oxidation were unaffected (Cyt and Oct, Figure 8).

A trend towards increased Complex IV-associated respiration was observed with the administration of RebA (Δ20.4%) (Figure 8). Nevertheless, the values are very scattered, and there are no significant differences between groups.

## 4. Discussion

This work showed that long-term (13 weeks) RebA consumption during the reproductive stage of female rats with the human ADI defined by EFSA did not majorly impact the cardiometabolic health of female rats.

Interestingly, RebA showed an atrophic effect in the heart without any repercussions on cardiac function. This result was further confirmed histologically, as demonstrated by a decreased cardiomyocyte cross-sectional area. Importantly, myocardial fibrosis was reduced in the RebA group, while mitochondrial and myofilamentary function was preserved. 

Glucose homeostasis was also not majorly affected by RebA consumption, except for a reduction in fasting glycemia at the end of the study, for values close to the lowest value of the rat’s euglycemic range (70–180 mg/dL) [13]. Sunanarunsawat and colleagues demonstrated increased insulin and decreased glucagon serum levels in diabetic rats, but not in the normoglycemic group, after receiving unspecified stevioside (0.25 g/kg body weight) or aqueous extract of *Stevia rebaudiana* (4.66 g/kg body weight) for 8 weeks [14]. These authors concluded that the beneficial impact of *Stevia rebaudiana* on glucagon and insulin levels could underlie its anti-hyperglycemic actions [14]. In our study, although the rats were normoglycemic, we detected a slight hypoglycemic trend induced by RebA as fasting glycemia decreased from 84.8 ± 3.04 mg/dL in the control group to 72.8 ± 2.18 mg/dL in the RebA group, in line with previous reports claiming that RebA reduces glycemia in humans [15,16,17] and in rats [18]. Moreover, the anti-hyperglycemic effects of stevia have been ascribed to the properties of stevioside and RebA as agonists of insulin receptors, causing an increase in glucose uptake by cardiac fibroblasts [19].

In our study, the observed decrease in liver weight could contribute to the observed hypoglycemic effect of RebA since the liver is the major source of glucose during fasting. Although other authors have shown no differences in liver weight, they report RebA (40 mg/Kg/day i.p.) to prevent liver injury, oxidative stress, and liver fibrosis caused by thioacetamide (200 mg/kg) [20]. In high-fat diet-induced nonalcoholic steatohepatitis obese mice, the administration of RebA decreased fasting glucose levels and hepatic stress enzymes, such as alanine aminotransferase (ALT) and plasma aspartate aminotransferase (AST) at 15 weeks post-intervention [21]. The authors suggest that hepatoprotection induced by RebA can be associated with, or even mediated by, improved pancreatic endocrine function that leads to anti-hyperglycemic effects [21]. Despite the observed decrease in in liver weight upon RebA treatment, we did not observe any changes in the hepatic ALT, AST, and ALP enzymes, suggesting that RebA at treatment dosage did not cause hepatic toxicity or hepatoprotection, probably because females were not submitted to a diet or metabolic challenge. Importantly, RebA doses used in the above-mentioned studies were higher than the dose used in the present study, which may also underlie variations in its biological effects. It is known that rat’s liver increases during pregnancy because of the proliferation of hepatocytes, a process that is reverted during weaning by hepatocytes’s death and liver remodeling [22]. As we could observe that RebA exposure decreased liver weight after 21 weeks, it would be interesting to evaluate liver changes during pregnancy and weaning to understand if this decrease in liver size observed only in dams exposed to RebA was already seen during pregnancy and weaning and to further understand the effects on offspring health.

RebA treatment also improved lipid homeostasis as it decreased total plasma cholesterol and LDL-c while maintaining the HDL-c/LDL-c ratio. Dyslipidemia is one of the major cardiovascular risk factors. In fact, total plasma cholesterol is directly associated with cardiovascular risk, as well as the decrease in HDL-c/LDL-c ratio. Although we could not observe an increase in the ratio HDL-c/LDL-c, the decrease in total plasma cholesterol upon RebA treatment already represents a better cardiovascular profile. Saravanan and Ramachandran also observed a decrease in total plasma cholesterol upon administration of a higher dose of RebA treatment (200 mg/kg/day for 45 days) on diabetic animals [23]. Ilias et al. suggested that RebA promotes the internalization of cholesterol by hepatocytes, modulating the expression of genes involved in cholesterol metabolism, such as *HMGCR*, *LDLR*, and *ACAT2* [24], which may, in part, explain our results. 

Despite the anti-diabetic actions, stevia derivatives are also reported to have anti-fibrotic effects [25], corroborating our results. Our study was carried out in healthy female rats. Pregnancy is expected to physiologically induce cardiac hypertrophy and decreased fibrosis, which are expected to revert in rodents 14 days after delivery [26], a process called postpartum cardiac reverse remodeling. In our study, RebA-fed females showed cardiac atrophy and decreased cardiac fibrosis far beyond the period of reverse remodeling (40 days after delivery). These alterations may underlie the observed LV atrophy.

The anti-fibrotic properties observed for Reb A are of great therapeutic potential and deserve to be further explored. Indeed, our data agree with the study by Zhao et al., who have shown that a non-identified stevioside reduced cardiac fibrosis in Type 1 diabetic mice by modulating the expression of matrix metallopeptidases while increasing glucose tolerance [27]. Conversely to our observations, stevioside increased heart weight but in diabetic animals, which, in the case of Zhao’s study, is beneficial as it counterbalances the type 1 diabetes-induced pathologic atrophy. It is important to note that, in that study, stevioside was used in a dose more than 10 times higher than the dose administered in our study.

To understand whether the morphological adaptations caused by RebA (reduced heart weight and decreased posterior and relative wall thicknesses) impacted cardiac function, we carried out in vivo echocardiographic and in vitro single-skinned cardiomyocyte evaluations. The observed structural changes did not affect cardiac myofilamentary function as assessed by preserved ejection fraction, cardiac output, and E/A between groups. Accordingly, in vitro, we did not find differences in cardiomyocyte contractile force, stiffness, or sensitivity for Ca^2+^ between groups. However, we cannot rule out that these results can be time- or dose-dependent. 

Known as critical metabolic players, mitochondria have been recognized as essential for multiple functions beyond energy production, regulating cell apoptosis, inflammation, and immunological processes [28,29]. It is also known that they are dysfunctional in several pathological conditions, such as obesity and diabetes [30]. Hepatic lipid accumulation may result from mitochondrial dysfunction [31], and several diabetic complications can be, in part, attributed to excessive production of ROS by mitochondria in a chronic hyperglycemia environment, increasing oxidative stress and activating stress-response pathways. Interestingly, RebA extended the lifespan and health of C. elegans via the reduction in ROS, decreasing aging lipid accumulation [32]. In addition to this, the microbiome–mitochondrion axis is currently suggested as an important player in cardiometabolic fitness [33,34], and RebA consumption has been described as a modulator of gut microbiota in obese dams [35] and in male rats [36].

Studies assessing the effect of *Stevia rebaudiana* (or their components, stevioside and rebaudioside) on mitochondrial function are scarce. To the best of our knowledge, this is the first study to explore the effects of steviol glycosides (RebA) on mitochondrial respiration using high-resolution respirometry. In accordance with metabolic and cardiac function results, we reported no differences in the function of cardiac mitochondria of female rats after 13 weeks of RebA treatment (4.71–5.61 mg steviol equivalents/kg body weight/day) when compared to the control group. However, a non-significant 20% increase in oxygen flux was observed in complex IV. Mutations and deficiencies in complex IV have been associated with dilated, hypertrophic, and histiocytoid cardiomyopathies [37]. In addition, in a model of insulin resistance with metabolic syndrome, myocardial mitochondria were smaller but had an increased amount of mitochondrial complex IV proteins reflecting an adaptative mitochondrial activity [38]. As we observed, RebA promoted a reduced heart weight, and the effect in Complex IV could be an adaptative response. Nonetheless, the results obtained are very scattered and non-significant in the RebA group.

Despite this, unravelling whether RebA chronic consumption could interfere with the microbiome–mitochondrion axis would be interesting.

In human breast cancer cells (MCF-7), stevioside is a potent inducer of apoptosis via altering mitochondrial transmembrane potential [39]. In rats supplemented with whey protein sweetened with 0.2% S. rebaudiana extract, peroxisome proliferator-activated receptor γ coactivator 1-α (PGC-1α), an important mitochondrial biogenesis marker, was significantly higher when compared with control animals [40]. In line with our results, Han and colleagues also demonstrated that mitochondrial activity was enhanced in the skeletal muscle of diabetic mice treated with a high dose of stevia extract (500 mg/kg/day) but not in the stevioside group (40 mg/kg/day) [41]. 

The physiological actions of RebA are still unclear due to the high heterogeneity of studies, the huge differences in doses used, the times of intervention, and stevia derivatives. In a systematic review with meta-analysis including 756 participants, RebA did not improve blood pressure or cardiovascular risk in contrast to stevioside, which seems to decrease blood pressure and fasting glycemia, although the small effect and the robustness of the results are limited to the heterogeneity of the studies included [42]. 

The fact that we did not find major cardiometabolic effects may be explained by the dose administrated, the one recommended by EFSA as safe for humans (4 mg/kg body weight/day). The more marked effects obtained in other studies described above may reflect the use of higher doses or combined with reducing glucose ingested. Considering that steviol glycoside is currently the third most common NSS across all food groups [43], new studies regarding exposures during the reproductive stage with higher doses of RebA are needed to analyze the effects of overconsumption of glucosylated steviol glycosides in comparison to the effects of recommended doses [9]. Although the absence of significant cardiometabolic alterations may indicate that this RebA dose is safe regarding female consumption during the reproductive stage, we did not evaluate these cardiometabolic parameters during pregnancy and lactation, which is a limitation of this study. The fact that RebA did not affect mating efficiency, gestational age at birth, litter size, or female-to-male ratio of each litter is also important regarding safety during the reproductive stage of the life cycle as they are indicators of successful reproduction.

Nevertheless, concerns have been raised regarding the safety of RebA consumption during pregnancy as it may pass the placenta [44] and induce metabolic disease in the offspring in the long term [35]. This issue deserves future research.

## 5. Conclusions

In conclusion, our work shows that chronic RebA consumption during the reproductive stage in the human dose recommended as safe by EFSA (4 mg/kg/day) induced a hypoglycemic effect without altering glucose tolerance or insulin sensitivity and improved blood lipid profile. Additionally, RebA consumption reduced cardiac fibrosis, preserving cardiac, myofilamentary, and mitochondrial function.

## Figures and Tables

**Figure 1 biology-13-00163-f001:**
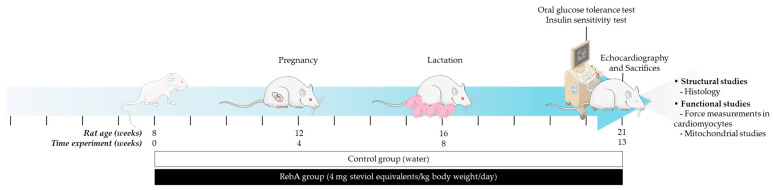
Experimental design. RebA, Rebaudioside A.

**Figure 2 biology-13-00163-f002:**
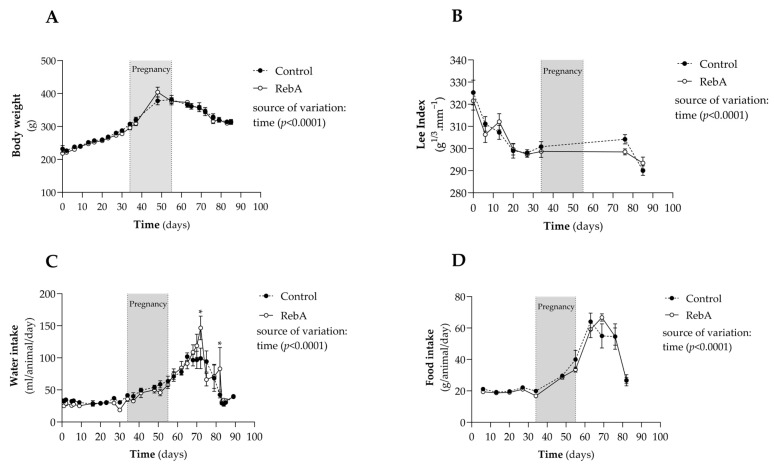
Body weight and feeding. Rebaudioside A in the drinking water (RebA) or drinking water alone (Control) was given from 4 weeks before mating (throughout pregnancy and lactation) until sacrifice at 21 weeks of age. During the study, (**A**) body weight, (**B**) Lee index, (**C**) water, and (**D**) food intake were measured. Results are presented as mean ± SEM (C or RebA, *n* = 8 each). Two-way ANOVA indicated a significant effect of time upon all measures (*p* < 0.001). * *p* < 0.05 versus Control (Bonferroni’s multiple comparisons test).

**Figure 3 biology-13-00163-f003:**
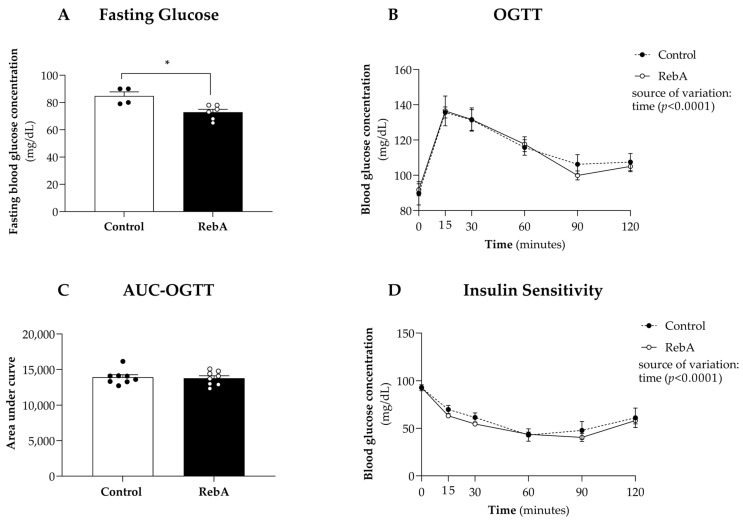
Glycemic control. Animals were treated with rebaudioside A in the drinking water (RebA) or with drinking water alone (Control) from 4 weeks before mating (throughout pregnancy and lactation) until sacrifice at 21 weeks of age. At 20 weeks of age, (**A**) fasting glycemia; (**B**) oral glucose tolerance test (OGTT); (**C**) the corresponding area under the curve (AUC); and (**D**) insulin sensitivity were measured. Data are represented as mean ± SEM (Control or RebA, 4 ≤ *n* ≤ 8 each). Two-way ANOVA indicated a significant effect of time upon all measures (*p* < 0.001). * *p* < 0.05 versus Control (*t*-test).

**Figure 4 biology-13-00163-f004:**
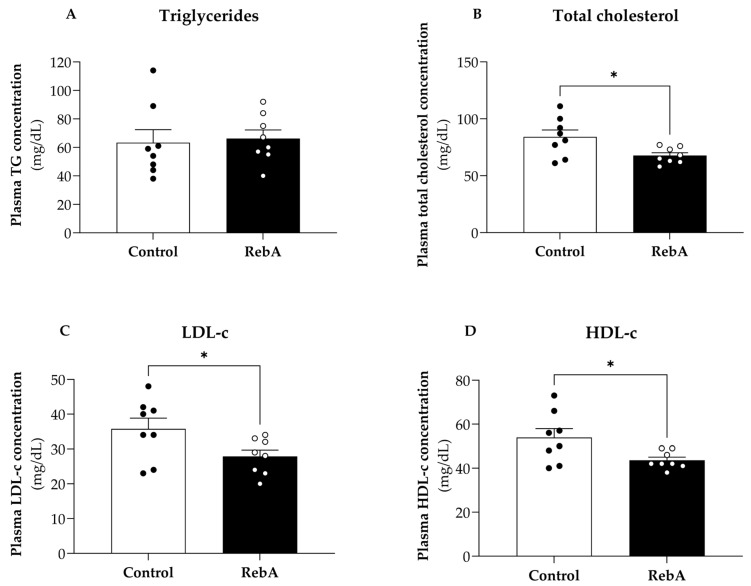
Lipid profile of animals at 21 weeks of age after treatment with rebaudioside A in the drinking water (RebA) or drinking water alone (Control) from 4 weeks before mating (throughout pregnancy and lactation) until sacrifice at 21 weeks of age. (**A**) Triglycerides, (**B**) total cholesterol, (**C**) low-density lipoprotein cholesterol (LDL-c), and (**D**) high-density lipoprotein cholesterol (HDL-c) were measured. Data are represented as mean ± SEM (Control or RebA, *n* = 8 each). T-test indicated a significant effect of RebA upon total cholesterol, LDL-c, and HDL-c. * *p* < 0.05 versus Control (*t*-test).

**Figure 5 biology-13-00163-f005:**
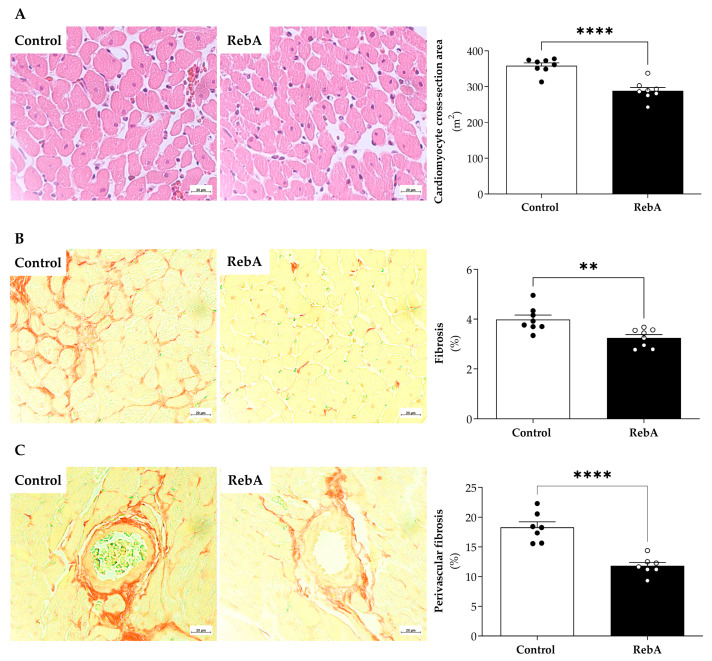
Cardiac morphological characterization at 21 weeks of age. Animals were treated from 4 weeks before mating (throughout pregnancy and lactation) until sacrifice (21 weeks of age) with rebaudioside A in the drinking water (RebA) or drinking water alone (Control). (**A**) Hematoxylin-eosin of left ventricular slices to assess cardiomyocyte cross-sectional area, (**B**) Picrosirius-red staining of left ventricular slices to assess fibrosis was measured, and (**C**) Picrosirius-red staining of left ventricular perivascular fibrosis. The representative sections of each group are represented at a magnification of ×40. Data are presented as mean ± SEM (Control or RebA, *n* = 8 each). T-test indicated a significant effect of RebA upon heart cardiomyocyte area, fibrosis, and perivascular fibrosis. ** *p* < 0.05 and **** *p* < 0.0001 versus Control (*t*-test).

**Figure 6 biology-13-00163-f006:**
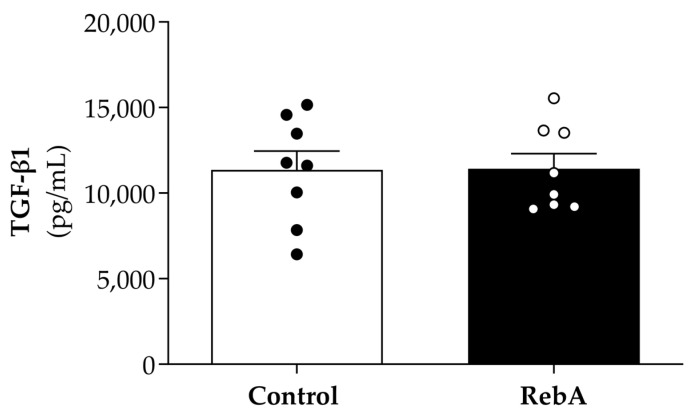
Plasma levels of TGF-β1 in animals treated from 4 weeks before mating (throughout pregnancy and lactation) until sacrifice (21 weeks of age) with rebaudioside A in the drinking water (RebA) or drinking water alone (Control).

**Figure 7 biology-13-00163-f007:**
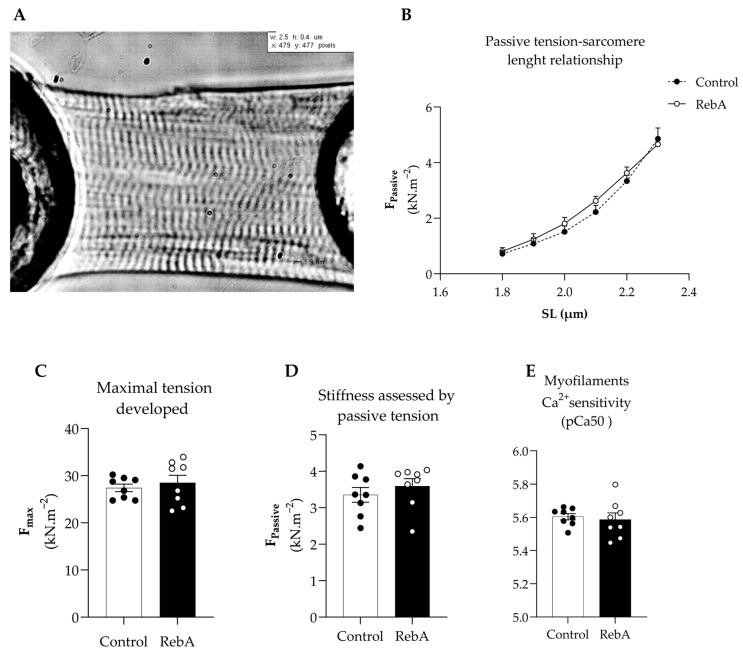
Force measurements in isolated permeabilized cardiomyocytes of the left ventricle (LV). Rebaudioside A in the drinking water (RebA) or drinking water alone (Control) was given from 4 weeks before mating (throughout pregnancy and lactation) until sacrifice at 21 weeks of age. (**A**) Representative image of a skinned cardiomyocyte stretched at 2.2 μm, (**B**) stiffness assessed by passive tension–sarcomere length relationship, (**C**) maximal tension developed, (**D**) stiffness assessed by passive tension, and (**E**) myofilament Ca^2+^-sensitivity expressed by pCa50. Data are presented as mean ± SEM (Control or RebA, *n* = 8 each).

**Figure 8 biology-13-00163-f008:**
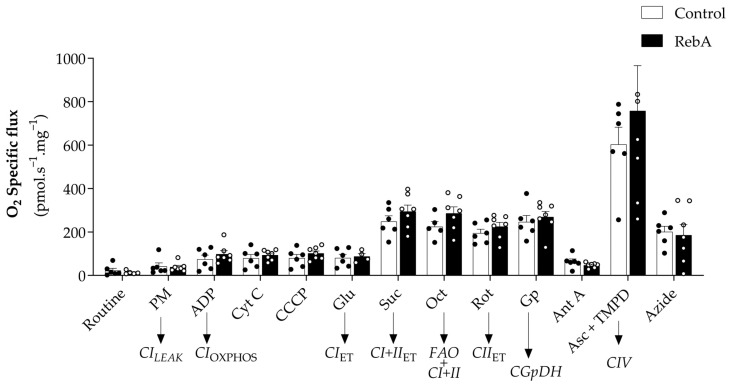
Heart mitochondrial function assessed by high-resolution respirometry, using the Oroboros Oxygraph-2k, of females treated with Rebaudioside A in the drinking water (RebA) or drinking water alone (Control) from 4 weeks before mating (throughout pregnancy and lactation) until sacrifice at 21 weeks of age. Results are presented as mean ± SEM (Control, *n* = 6 and RebA, *n* = 7). (T-test or Mann–Whitney test). PM: pyruvate + malate (Complex I, leak state); ADP: adenosine diphosphate (Complex I, OXPHOS state); Cyt C: cytochrome c (test for mitochondrial outer membrane integrity); CCCP (uncoupler); Glu: glutamate (Complex I, electron transfer (ET) state); Suc: succinate (Complex I+II, ET state); Oct: octanoylcarnitine (Fatty acid oxidation + Complex I+II, ET state); Rot: rotenone (Complex II, ET state); Gp: glycerophosphate (Complex II + glycerophosphate dehydrogenase, ET state); Ant A: antimycin A (residual oxygen consumption, ROX); Asc + TMPD: ascorbate + TMPD (Complex IV); Azide: sodium azide.

**Table 1 biology-13-00163-t001:** Gestational and morphometric data.

	Control				RebA				*p*-Value ^b^
	Mean	±	SEM	(*n*)	Mean	±	SEM	(*n*)	
Mating efficiency (*n* mating attempts) ^a^	1.9	±	0.4	(8)	2.5	±	0.5	(8)	0.300
Gestational age at birth (days)	21.5	±	0.3	(8)	21.9	±	0.1	(8)	0.224
Litter size (pups)	12.5	±	1.1	(8)	14.3	±	0.5	(8)	0.526
Female-to-male ratio	1.1	±	0.2	(6)	0.8	±	0.1	(8)	0.138
Body weight at euthanasia (g)	323.4	±	4.8	(8)	314.5	±	5.8	(8)	0.257

^a^ Corresponds to the number of encounters with males until conception was confirmed. ^b^
*t*-test. RebA, rebaudioside A; SEM, standard error of the mean.

**Table 2 biology-13-00163-t002:** Organ weight adjusted for body surface area.

Organ Weight	Control				RebA				*p*-Value ^b^
	Mean	±	SEM	(*n*)	Mean	±	SEM	(*n*)	
Liver/BSA (mg/cm^2^)	23.3	±	0.38	(8)	21.6	±	0.31	(8)	0.005
Pancreas/BSA (mg/cm^2^)	1.94	±	0.22	(7)	2.42	±	0.21	(8)	0.188
Heart/BSA (mg/cm^2^)	26.8	±	0.05	(8)	25.0	±	0.04	(8)	0.031
Skeletal muscle/BSA (mg/cm^2^)	4.73	±	0.17	(8)	4.87	±	0.06	(7)	0.505

^b^ *t*-test; RebA, rebaudioside A; SEM, standard error of the mean.

**Table 3 biology-13-00163-t003:** In vivo echocardiographic evaluation of cardiac function and structure.

Parameter	Control Mean ± SEM	(*n*)	RebA Mean ± SEM	(*n*)	*p*-Value
Heart rate (bpm)	287.5 ± 6.9	(8)	269.2 ± 9.4	(8)	0.165
LV end-diastolic volume/BSA (cm^3^.cm^−2^)	1.260 ± 0.065	(8)	1.20 ± 0.054	(8)	0.604
LV end-systolic volume/BSA (cm^3^.cm^−2^)	0.049 ± 0.042	(8)	0.050 ± 0.040	(8)	0.969
Interventricular septum (mm)	1.48 ± 0.13	(8)	1.39 ± 0.13	(8)	0.171
Posterior LV wall (mm)	0.004 ± 85 × 10^−5^	(8)	0.003 ± 12 × 10^−5^	(8)	0.047
LV mass/BSA (g·cm^−2^)	1.614 ± 0.081	(8)	1.508 ± 0.060	(8)	0.329
Relative wall thickness	0.890 ± 0.025	(8)	0.780 ± 0.029	(8)	0.025
Ejection fraction (%)	75.38 ± 2.30	(8)	71.28 ± 1.43	(8)	0.176
Cardiac index mL·min·cm^−2^	0.113 ± 0.005	(8)	0.102 ± 0.005	(8)	0.152
E/A	1.88 ± 0.54	(8)	2.15 ± 0.69	(8)	0.398
Tei index	0.44 ± 0.06	(8)	0.46 ± 0.10	(8)	0.748

RebA, rebaudioside A; SEM, standard error of the mean; BSA, body surface area; E/A: ratio between peak E and A waves of pulsed-wave Doppler mitral flow velocity; LV, left ventricle.

## Data Availability

Data are contained within the article.

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
