# Peer review of "The Cardiometabolic Impact of Rebaudioside A Exposure during the Reproductive Stage"

_biology, 2024, doi:10.3390/biology13030163_

Round 1
Reviewer 1 Report
Comments and Suggestions for Authors
The primary objective of the authors was to explore the prolonged effects of RebA consumption on mitochondrial and cardiometabolic parameters during the reproductive stage of the female rat life cycle. Female rats exposed to RebA (4 mg/kg body weight/day of steviol glycosides) in their drinking water, prior to mating until the weaning, were assessed for food and water intake, glucose and lipid balance, heart structure, function, and mitochondrial performance. RebA reduced heart size at 21-week analysis, leading to a reduction in cardiomyocyte cross-sectional area and myocardial fibrosis, while cardiac function remained unaffected. At 21-weeks, no notable changes were observed in mitochondrial and myofilament functions. Glucose tolerance and insulin sensitivity were not affected, but fasting glycemia and total plasma cholesterol were decreased.
Though the authors claim that this study demonstrated that prolonged RebA consumption during the reproductive stage of the female rat life cycle, at doses close to EFSA's defined acceptable daily intake (ADI), did not significantly impact the cardiometabolic health of female rats, the results should be interpreted with caution as cardiometabolic parameters were assessed only at 21 weeks and there are no measurements for those parameters during pregnancy and lactation, and its long-term effect on the offspring. As this issue focuses on mitochondria, it would be insightful for the authors to rigorously test mitochondrial dynamics and energetics, and test the link between gut microbiota alteration and mitochondrial function upon RebA consumption.
SPECIFIC COMMENTS:
1. The researchers examined the post-weaning period to evaluate the enduring effects of RebA, utilizing the dosage recommended by EFSA as safe for pregnant individuals. Although it can be inferred that prolonged RebA consumption does not have adverse effects on the mother in the long term after childbirth, it cannot be definitively established that RebA consumption is safe during pregnancy. This uncertainty arises due to the lack of measurement of various crucial parameters during the pregnancy and lactation periods. In their conclusion and interpretation of data, authors should clearly mention this caveat.
2. What is the duration of residence for RebA and its hydrolyzed product, steviol, in the rat body? Although RebA can be metabolized within 24 hours, steviol persists for several days before being cleared. Are RebA or steviol detectable in the blood, and if so, what are their respective half-lives?
3. During pregnancy, an elevation in liver weight has been noted, surpassing the anticipated increase solely attributable to overall weight gain. Female rat livers exhibited hepatocyte proliferation through an unrecognized developmental program mediated by pregnancy. Considering RebA's capacity to reduce liver size after 21 weeks, it would have been valuable to monitor changes during pregnancy to understand its effects on liver function and size, as these factors influence the well-being of the litter.
4. Throughout pregnancy, the heart undergoes substantial cardiac remodeling, experiencing approximately a 50% increase in left ventricular (LV) mass to accommodate the additional blood volume required. Since RebA has been observed to diminish heart size, particularly affecting the posterior LV wall and relative wall thickness, it would be insightful to investigate how RebA influences the trajectory of cardiac remodeling during pregnancy and its impact on various cardiometabolic markers in this specific context.
5. The authors should test and ascertain if RebA impacts gut microbiota. Given that steviol, known for its prolonged presence, can alter gut microbiota, is there a similar effect observed with RebA?
Considering the established fact that NSS can alter gut microbiota, subsequently influencing mitochondrial metabolism (https://www.ncbi.nlm.nih.gov/pmc/articles/PMC5425687/), it would be beneficial to investigate alterations in gut microbiota during RebA consumption, particularly in pregnancy, using fecal samples. Additionally, examining changes in short-chain fatty acids could provide valuable insights into the correlation between gut microbiota changes and mitochondrial metabolism. Maternal NSS consumption fed obesogenic diet impacted metabolic outcomes, gut microbiota and altered gene expression in the mesolimbic reward system in dams and the offsprings (10.1136/gutjnl-2018-317505).
6. Regarding the effect of RebA on mitochondria, as this issue is especially focused on mitochondria, was there any alteration in the rate of oxygen consumption? Is it plausible that RebA influenced the mitochondrial transmembrane potential?
7. Does RebA exert any impact on cytosolic Ca2+, a recognized regulator of mitochondrial energetics and a determinant of cardiac power output through thin filament activation? The use of Fura-2 AM dye could facilitate the observation of changes in Ca2+ flux associated with RebA consumption.
8. The authors should make it clear whether pure RebA was administered or steviol glycosides were administered? If only RebA was administered, its purity should be mentioned.
9. The source of all chemical reagents used hasn’t been mentioned in material and methods section.
10. Does the glucose tolerance change between control and RebA group during pregnancy, which is well known to remodel metabolism?
11. The authors should comment and elaborate on the 20% increase in Complex IV associated respiration with RebA consumption and its effect on the metabolic output.
12. Fig 6 (E), the Y-axis needs to be changed to pCa or Ca2+ concentration. It currently reads kNm-2
13. RebA consumption could impact offsprings and the authors point out this caveat with the study. Were the offsprings tested for RebA to find if RebA and/or steviol passes through placenta into the fetus? Also, is RebA and/or steviol detectable in the breast milk of the female rats?
Comments on the Quality of English LanguageThe English language is comprehensible, but the manuscript requires proofreading and editing to rectify minor spelling errors. For example, Myofilament has been wrongly written as miofilament at multiple places in the manuscript.
Author Response
We greatly appreciate the insightful comments from the Reviewers and the Editor on the manuscript which contributed to clarify several points and improve the quality of the manuscript. We have incorporated changes to reflect the suggestions provided and have highlighted the changes in yellow throughout the manuscript. In addition, all the changes are described in each response. Language imperfections were corrected throughout, mainly in the discussion section.

Reviewer 2 Report
Comments and Suggestions for Authors
1. For the dose of RebaudiosideA, rats were treated with RebaudiosideA (4 mg/kg body weight/day of steviol glycosides). You used the same human dose of 4 mg/kg body weight/day, however when converting the human dose to animal dose according to this equation:
AED (mg / kg) = Human does (mg / kg) × Km ratio, it should be 24.6 mg/kg meaning the animal dose should be 6 times more than the dose used in this study. the reference for this equation is attached. So, please clarify that.
Nair AB, Jacob S. A simple practice guide for dose conversion between animals and human. J Basic Clin Pharm. 2016 Mar;7(2):27-31. doi: 10.4103/0976-0105.177703. PMID: 27057123; PMCID: PMC4804402. (reference for dose conversion)
2. Figure 5A, instead of H & E staining, WGA staining would give more appropriate information of the atrophic or hypertrophic nature of myocytes.
3. Did the authors observe any differences in the heart vasculature?
4. It is recommended to confirm the reduced cardiac fibrosis by measuring the expression levels of fibrotic markers.
5. In Figure 5, please include the magnification of the images in the figure caption.
6. In the results, you mentioned that RebA fed rats showed decreased cardiomyocyte cross-section area and cardiac atrophy. However, in the discussion, you mentioned that RebA fed rats showed hypertrophy, please explain.
Comments on the Quality of English Language
Minor editing of English language is required
Author Response

(The authors gave the same response as above.)
